Putative fossil blood cells reinterpreted as diagenetic structures

Korneisel Dana E. danak@vt.edu 1
Nesbitt Sterling J. 1
Werning Sarah 2
Xiao Shuhai 1
1 Department of Geosciences, Virginia Polytechnic Institute and State University (Virginia Tech) , Blacksburg , VA , United States of America
2 Department of Anatomy, Des Moines University , Des Moines , IA , United States of America
Lieberman Bruce
Electronic publication date: 2021 Dec 16
Publication date: 2021
Volume: 9
Electronic Location ID: e12651
Received 2021 Sep 12; Accepted 2021 Nov 28
Copyright: ©2021 Korneisel et al.
Copyright year: 2021
Copyright holder: Korneisel et al.
License: This is an open access article distributed under the terms of the Creative Commons Attribution License, which permits unrestricted use, distribution, reproduction and adaptation in any medium and for any purpose provided that it is properly attributed. For attribution, the original author(s), title, publication source (PeerJ) and either DOI or URL of the article must be cited.
License URL: https://creativecommons.org/licenses/by/4.0/

Keywords: Taphonomy, Red blood cell, Preservation, Therizinosaur, Fossil blood, Histology, TOF-SIMS, EDS, Raman spectroscopy, Cretaceous

Funding: NASA Exobiology grant to SX 80NSSC18K1086 GSA graduate student research grant to DEK This research was supported by a NASA Exobiology grant to SX (award number 80NSSC18K1086) and a GSA graduate student research grant to DEK. There was no additional external funding received for this study. The funders had no role in study design, data collection and analysis, decision to publish, or preparation of the manuscript.

==============================
Red to red-orange spheres in the vascular canals of fossil bone thin sections have been repeatedly reported using light microscopy. Some of these have been interpreted as the fossilized remains of blood cells or, alternatively, pyrite framboids. Here, we assess claims of blood cell preservation within bones of the therizinosauroid theropod Beipiaosaurus inexpectus from the Jehol Lagerstätte. Using Raman spectroscopy, Energy Dispersive X-ray Spectrometry, and Time of Flight Secondary Ion Mass Spectroscopy, we found evidence of high taphonomic alteration of the bone. We also found that the vascular canals in the bone, once purported to contain fossil red blood cell, are filled with a mix of clay minerals and carbonaceous compounds. The spheres could not be analyzed in isolation, but we did not find any evidence of pyrite or heme compounds in the vessels, surrounding bone, or matrix. However, we did observe similar spheres under light microscopy in petrified wood found in proximity to the dinosaur. Consequently, we conclude that the red spheres are most likely diagenetic structures replicated by the clay minerals present throughout the vascular canals.

Introduction

The Jehol Biota of the Yixian Formation in Northeast China is a Lagerstätte (Muscente et al., 2017; Pan, Sha & Fuersich, 2014; Zhou, 2014), an exceptional site both in the concentration of fossils and in the quality of preservation (Chang, 2011; Pan et al., 2013). The lower Cretaceous lacustrine sediments that host this biota are famous for preserving abundant avialians (Chinsamy et al., 2013; Hou et al., 1995; Zhou, 2006a; Zhou, 2006b) with epidermal outlines including feathers (Ji et al., 2007; Li et al., 2012; Wang, Dong & Evans, 2010; Xu et al., 2003; Xu, Zheng & You, 2009; Yuan, 2000) and a rich array of other vertebrates with soft tissue preservation (Chang et al., 2003). Molecular preservation, including melanosomes with intact melanin, has been observed in fossilized eyes, hairs, and feathers (Pan et al., 2016; Wogelius et al., 2011) preserved next to bones with fine surficial and histological details (e.g., canaliculi) (O’Connor et al., 2014; Wang et al., 2019; Yao, Zhang & Tang, 2002). The specimen examined herein, the holotype (IVPP V11559) of Beipiaosaurus inexpectus, has fine surface preservation of the bone, feather preservation, and has been proposed to contain fossilized red blood cells based on the appearance of red spherical structures in the bone’s vascular canals under light microscopy (Yao, Zhang & Tang, 2002).

Structures similar to those interpreted as red blood cells in B. inexpectus have also been identified as erythrocytes in other fossils since 1907 (Seitz, 1907), but the fossilization potential of blood cells became a popular subject at the end of the 20th and into the 21st century (e.g., Pawlicki & Nowogrodzka-Zagórska, 1998; Schweitzer, Wittmeyer & Horner, 2007; Wilby, 1993; Yao, Zhang & Tang, 2002). A handful of studies explored the idea of fossilizing blood cells more deeply (Kaye, Gaugler & Sawlowicz, 2008; Martill & Unwin, 1997; Schweitzer & Horner, 1999), and some incorporated data on blood cell shape, chemistry, and the effects of desiccation when identifying microstructures in a fossil as blood cells (Bertazzo et al., 2015; Pawlicki & Nowogrodzka-Zagórska, 1998). Most hypothesized fossil red blood cells, however, have not been thoroughly analyzed using chemical tools (Moodie, 1920; Pawlicki & Nowogrodzka-Zagórska, 1998; Plet et al., 2017; Seitz, 1907; Wilby, 1993; Yao, Zhang & Tang, 2002).

Here, we analyze the preservation and vascular canal contents of the holotype of B. inexpectus. We aim to determine the composition of this specimen’s putative blood cells and the degree of diagenetic alteration of the specimen using the following combination of analytical tools: light microscopy, Scanning Electron Microscopy (SEM), Energy Dispersive X-ray Spectroscopy (EDS or EDX), Raman spectroscopy (Raman), and Time of Flight Secondary Ion Mass Spectrometry (TOF-SIMS). Although each analytical tool has limitations, the combined strengths of these tools offer new insights into intra-vascular preservation.

Geological setting

Beipiaosaurus inexpectus (IVPP V11559) comes from the Sihetun locality of the Yixian Formation. The locality is a protected site within the Beipiao Bird Fossil National Nature Reserve of Liaoning Province, China (Figs. 1A, 1B) (Wang, 1998; Xu, Tang & Wang, 1999).

Figure 1 Geographic and stratigraphic setting of Beipiaosaurus inexpectus (IVPP V11559).

(A) Map of China with box denoting the area enlarged in (B). (B) The Sihetun locality (black square) in Liaoning Province. (C) Stratigraphic column of the Yixian Formation at Sihetun, showing the stratigraphic horizon of the holotype specimen of Beipiaosaurus inexpectus (marked as “D”). Redrawn from Yao, Zhang & Tang (2002) and field observations. This section is approximately 15 m from the original collection site and is now covered by new construction. Tuff thicknesses vary laterally. (D) Detailed stratigraphy of the fossil horizon of the holotype specimen of Beipiaosaurus inexpectus. Drawn from field observations.

In the Sihetun area, the Yixian is composed of two sedimentary and two volcanic units. The Lujiatun Unit is the lowest and consists of coarse-grained sediments. This is succeeded by the lower volcanic unit, the shales and siltstones of the Jianshangou Unit, and then the upper volcanic unit (Jiang et al., 2011; Jiang & Sha, 2007). The tuffaceous claystones and shales of the Jianshangou Unit host B. inexpectus as well as many other vertebrate and invertebrate fossils (Chang, 2011; Jiang & Sha, 2007; Pan et al., 2013; Zhou, 2006b). Radiometrically dated (40Ar/39Ar) tuffs near the fossil-bearing layer constrain this unit to 124 Ma (Chang et al., 2009)—125 Ma (Fig. 1C) (Swisher et al., 2002), consistent with palynological data (Li & Batten, 2007). The Jianshangou Unit is bracketed by the volcanic units, dated at 125–128 Ma (lower volcanic unit) (Chang et al., 2017) and approximately 122 Ma (upper volcanic unit) (Jiang et al., 2011; Wang et al., 2001). At the Sihetun locality, the layer bearing B. inexpectus (beds 25–29 in Wang et al., 1999) underlies a tuff dated between 124.35 and 126.1 Ma by less than 3.5 m (Fig. 1C) (Wang et al., 1999). Based on these dates, the specimen of focus in this study is from the late Barremian or early Aptian age of the Early Cretaceous Period (Chang et al., 2017; Chang et al., 2009; Cohen et al., 2013 (updated); He et al., 2006; Jiang & Sha, 2007; Wang et al., 2001).

Materials & Methods

All samples examined in this study, as well as slide/sample number, element (if applicable), and analyses performed, are listed in Table 1.

Table 1 Materials used in this study.

All dinosaur material is from IVPP V11559 (holotype specimen of Beipiaosaurus inexpectus). Each specimen listed here was examined during this study, and its anatomical and preparatory context are listed in addition to the types of analyses used. Some were not part of the final results.

Specimen	Source	Type of sample	Element	Position	Orientation	Analyses conducted	Locations analyzed, Raman	
2018-X1	Prepared by authors from materials provided by Yao, et al.	Thin Section	Gastralium	near broken end of 2 cm fragment	Transverse	histology	N/A	
2018-X2	Prepared by authors from materials provided by Yao, et al.	Thin Section	Gastralium	serial section from 2018-X1	Transverse	histology	N/A	
2018-L1	Prepared by authors from materials provided by Yao, et al.	Thin Section	Gastralium	exterior, serial section with 2018-L2-5	Longitudinal	histology, Raman, EDS	bone, epoxy, plexiglass, glue, various vessel fills, spheres, lacunae	
2018-L2	Prepared by authors from materials provided by Yao, et al.	Thin Section	Gastralium	exterior, serial section with 2018-L2-5	Longitudinal	histology, Raman, EDS	bone, epoxy, various vessel fills, spheres, lacunae	
2018-L3	Prepared by authors from materials provided by Yao, et al.	Thin Section	Gastralium	exterior, serial section with 2018-L2-5	Longitudinal	histology, Raman	various vessel fills, spheres	
2018-L4	Prepared by authors from materials provided by Yao, et al.	Thin Section	Gastralium	exterior, serial section with 2018-L2-5	Longitudinal	histology, Raman	various vessel fills, spheres	
2018-L5	Prepared by authors from materials provided by Yao, et al.	Thin Section	Gastralium	exterior, serial section with 2018-L2-5	Longitudinal	histology, Raman	various vessel fills, spheres	
HO-9601	Prepared by Yao, et al.	Thin Section	Humerus	unknown	Longitudinal	histology	N/A	
HO-9602	Prepared by Yao, et al.	Thin Section	Humerus	shaft	Transverse	histology	N/A	
LJ98B-1	Prepared by Yao, et al.	Thin Section	Humerus	unknown	Longitudinal	histology	N/A	
LJ98B-4	Prepared by Yao, et al.	Thin Section	Humerus	shaft	Transverse	histology	N/A	
2018-1	Prepared by authors from materials provided by Yao, et al.	Thin Section	Petrified Wood	N/A	para-transverse	histology	N/A	
2018-X3	Prepared by authors from materials provided by Xu	Thick Section from Hand Sample	Gastralium	near break, central in element	Transverse	TOF-SIMS	N/A	
Alligator	Provided by Stocker	Unprepared bone	Rib	exterior	N/A	Raman	bone matrix	

For histological analysis of fossil bone under light microscopy, we re-analyzed the petrographic thin sections of the left humerus of the B. inexpectus holotype (IVPP V11559) described in Yao, Zhang & Tang (2002) (see Fig. 2A for sampling locations). The original thin sections by Yao, Zhang & Tang (2002) were covered in a highly fluorescent epoxy as well as a glass coverslip, and thus were not useful for Raman, EDS, SEM, or TOF-SIMS analysis.

Figure 2 Photograph of sampled specimen (Beipiaosaurus inexpectus, IVPP V11559) (A) and transmitted light micrographs of representative thin sections (B–D).

In thin section images, black arrows indicate spheres (putative red blood cells), white arrows indicate osteocyte lacunae, and grey arrows indicate non-spherical vessel fills (A) Initially published portion of IVPP V11559, with sampled areas marked “1” and “2”. Samples from these areas include previously published thin sections (Yao, Zhang & Tang, 2002) as well as thin sections produced by the authors for this study. We also sampled fragments of associated gastralia (not pictured here) (B) Vessel containing spheres as well as non-spherical amorphous vessel fill (grey bracket) in thin section LJ98B-1 (sample area 1; originally prepared for (Yao, Zhang & Tang, 2002)). (C) Spheres in an anastomosing vessel in thin section 2018-L2 (newly prepared thin section from sample area 2). (D) Small grainy vessel fill, also from thin section LJ98B-1.

For SEM, Raman, and EDS analysis (as well as additional light microscopy examination of histology), we prepared new petrographic thin sections from materials collected during the initial excavation of IVPP V11559 (Yao, Zhang & Tang, 2002). These include a fragmentary gastralium and additional fragments of the left humerus from the holotype, as well as petrified wood found in association with this individual. A hand sample containing gastralia and associated matrix from the holotype specimen was sampled directly for TOF-SIMS analysis. Finally, we also prepared thin sections of modern bone for comparison of Raman spectra.

a. Petrographic sectioning

To prepare thin sections of B. inexpectus material and petrified wood, we embedded the fragments in Castolite-AC (EP4101polyester casting resin; Eager Polymers, Chicago, IL) catalyzed with MEKP, and then cut 0.5 mm thick slices on an IsoMet 1000 Precision Sectioning Saw (Buehler, Lake Bluff, IL) using tap water to fill the basin (for lubrication) and approximately 5 mL of Cool 2 Cutting Fluid (for cooling; Buehler). Before processing each new specimen, we cleaned and dressed the blade using a priming block. We air-dried slices for a minimum of 12 h. Before mounting, we roughened slice-sized areas on mounting side of plexiglass slides with a 120-grit sanding sponge to improve specimen adhesion. We mounted each slice on a roughened plexiglass slide using Loctite ULTRA Gel Control commercial superglue (Henkel Corp., Bridgewater, NJ). Mounted slices were then ground on a Metaserv 2000 Grinder/Polisher (Buehler) until they were thin enough to view details in transmitted light (approximately 100 µm thick). We first ground the specimens using 240 grit until they were ∼300 µm thick, then 400 grit to ∼150 µm, and finally using 800 grit until the specimens were transparent enough that their histological features were visible when dry. All specimens prepared for this study were rinsed clean, but their surfaces received no further chemical treatment after grinding (unless otherwise noted), and they were not coverslipped.

b. Raman spectroscopy

Raman spectroscopy is a well-established method in taphonomy (Bernard et al., 2007; Jehlička, Jorge Villar & Edwards, 2004; Marshall et al., 2012; Thomas et al., 2007; Thomas et al., 2011; Wiemann, Yang & Norell, 2018; Witke et al., 2004). It is useful for assessing the degree of diagenetic alteration to bone (Thomas et al., 2007; Thomas et al., 2011) and evaluating the origin of compounds and structures in fossil material (Marshall et al., 2012; Thomas et al., 2014; Wiemann, Yang & Norell, 2018).

We employed Raman spectroscopy to test the alternative interpretation of the putative red blood cells as framboids of iron minerals (Martill & Unwin, 1997), and to assess the quality of bone preservation from a diagenetic perspective. All Raman spectra were obtained from thin sectioned material, and collected on a high-resolution, 800 mm focal length spectrometer (LabRAM HR800; Horiba Scientific, Kyoto, Japan) with a 785 nm diode laser at the Virginia Tech Vibrational Spectroscopy Lab. The laser beam was focused on an approximately 8 µm diameter spot to reduce heat on the sample and points for analysis were chosen using a 50x objective lens. Raman spectral data were collected at 1/10 power (maximum power = 150 mW) for 5 s per collection. Decreased laser power reduces the intensity of Raman peaks, but also lessens laser damage of the specimen. As blank tests, we also took Raman spectra of the plexiglass, epoxy, and superglue used in slide preparation. The relatively thick (100 µm) sections and the weak laser power used in the analysis ensured that the Raman data were not contaminated by epoxy or plexiglass signals; this was confirmed by a comparison between sample data and blank data. The spectral resolution is 0.69 wave numbers. Raman spectra were baselined using the Gaussian baseline correction algorithm in Fityk (Wojdyr, 2010) and CrystalSleuth (Laetsch & Downs, 2006). Reference peak positions (e.g., for pyrite and apatite) were obtained from the CrystalSleuth database.

c. Energy dispersive X-ray spectrometry

To locate vessels exposed on the surface of thin sections and to obtain element maps of vessel-filling material and near-vessel bone, we used a Hitachi TM-3000 Tabletop SEM coupled with a Quantax 70 Energy Dispersive X-ray Spectrophotometer (EDS) system (Hitachi High-Tech America, Inc., Schaumburg, IL). In initial analyses, we left the samples uncoated, but surrounded them with aluminum tape to improve conductivity, and used the Quantax70 to interpret and visualize the EDS data. Based on these preliminary analyses, we identified specimen 2018-L2 (longitudinal section of a gastralium fragment, see Table 1) as the best candidate for quantitative EDS spot analysis due to the abundance of vessels exposed on the surface of the section.

We then collected spot analyses from 2018-L2 on an FEI Quanta 600FEG environmental SEM (FEI Company, Hillsboro, OR) with both back scattered electron (BSE) and secondary electron (SE) capability, operating at a voltage of 5–20 kV. The sample was coated in a mixture of gold and palladium and surrounded by aluminum tape to improve conductivity. We used a Bruker EDX to carry out spot analyses of elemental concentrations and to generate additional elemental maps on vessel fills and lacuna fills.

d. Time of flight - secondary ion mass spectrometry

Relative to other mass spectrometers for fossil analysis, TOF-SIMS is minimally destructive; it removes only about 1 nm of surface material and charges material up to 30 nm deep in the specimen, a depth easily removed with sputtering if further analysis in a spot is desired (Cheng, Wucher & Winograd, 2006; Debois, Brunelle & Laprévote, 2007; Touboul et al., 2005). TOF-SIMS creates a map of relative ion abundance across an area of a specimen, can analyze both organic and inorganic molecules simultaneously, and can detect molecules as well as individual elements with superb spatial precision (Thiel & Sjövall, 2014).

To minimize potential contamination from epoxy, we did not embed TOF-SIMS specimens. Instead, we used a hand sample of gastralia from IVPP V11559, which is surrounded by finely laminated mudstone matrix. We ground a fresh edge of a transverse break through one gastralium (section 2018-X3; see Table 1) using 240 grit and then 800 grit sandpaper on the Metaserv 2000 Grinder/Polisher described above. Just prior to TOF-SIMS analysis, a slice approximately 1 mm thick (including the ground edge) was cut off with a Dremel tool and rinsed with isopropyl alcohol in order to remove any surface contamination from handling the specimen during collection, storage, and processing. Although isopropyl may remove carbonaceous compounds, this step was necessary because IVPP V11559 was not originally collected and handled with chemical analyses in mind. Section 2018-X3 was then mounted on a silicon stub and placed into the vacuum chamber of an IONTOF TOF.SIMS 5 (Iontof GmbH, Munster, Germany). Areas of interest were sputtered with a 2 kV cesium beam for a minimum of 5 min to reveal a clean surface at depth (Graham & Gamble, 2018; Taylor, Graham & Castner, 2015). Spectra were acquired from the sputtered surfaces using a 30 kV bismuth ion beam (primary species Bi3+), as bismuth is not of biological interest and Bi3+ is very precise (Touboul et al., 2005).

We calibrated for mass using C, 18O, O2, F, Na, C2, Al, Si, P, Cl, 37Cl, PO, PO2, and PO3. After identifying calibration peaks, we defined the width of each peak and then assigned non-calibration peaks starting at low masses. We aimed for deviation measures (the difference of an observed peak from the expected location for a given secondary ion) below 200 ppm to improve peak assignment accuracy. Components of shouldered peaks (see (Green, Gilmore & Seah, 2006) for comparable peak morphology) were distinguished from one another whenever the IONTOF TOF.SIMS 5 software was able to separate them, and we placed the boundary at the nadir of the valley or inflection point of the shoulder.

Results

a. Light microscopy

Sections 2018-L1 through 2018-L5 (a series of longitudinal thin sections from a fragment of a gastralium, see Table 1) are roughly rectangular (2C & 3A–3D). The bone growth pattern does not differ substantially in the exterior and interior of the bone, which is expected for gastralia (e.g., Klein, Canoville & Houssaye, 2019).

Figure 3 Raman spectra collected from thin sections 2018-L1–2018-L5 (gastralia fragment from Beipiaosaurus inexpectus, IVPP V11559) and from a modern alligator bone sample.

Vertical bands represent approximate locations of the characteristic Raman peaks of pyrite, magnetite, and apatite. (A–D) Examples of sample locations (blue dots) targeted for Raman spectroscopic analysis. These are not necessarily the spot analyzed, but serve to illustrate the type of morphological features selected for Raman analysis. (A’–D’) Raman spectra corresponding to morphological features marked by blue dots in (A–D). These spectra correspond to “2018_L2_784_pos2_indredorange2”, “2018_L2_pos2_bone”, “2018_L2_784_pos2_redfill”, and “2018_L2_ 784_pos2_round4” in the supplementary data, respectively. Regardless of the morphological feature analyzed in this fossil, the Raman spectra show six peaks, with average wavenumbers of 962.4, 1,212, 1,296, 1,430, 1,570, and 1,690 cm−1. (E’) Raman spectrum of a modern alligator bone. Apatite is apparent throughout this thin section, whereas iron-bearing minerals are not present.

Sections 2018-L1 through 5 visually resemble sections LJ98B-1 & LJ98B-4 from Yao, Zhang & Tang (2002) in terms of their vascularization pattern, lacunae with preserved canaliculi, and a diversity of vessel fills. Most vascular canals are longitudinally oriented, but they anastomose laterally and through the depth of the section, connecting otherwise parallel longitudinal canals. Osteocyte lacunae are visible in each of these sections and are generally lenticular in shape (Figs. 2B–2D & 3A–3D). They are densely distributed throughout the sections (approximately 650/mm2) except at the bone margin and directly adjacent to vascular canals. Canaliculi are often visible and the sections have a high visual quality, similar to a modern histological section (Hedges, Millard & Pike, 1995). Vessel fills have a variety of textures and colors, from massive and opaque black to granular and translucent orange (Figs. 3A–3D). These fills include translucent, red-orange spheres located throughout the vascular canals.

The spheres (i.e., purported red blood cells) are most abundant at points where two vascular canals connect (Fig. 3D) and near other shapes and textures of vessel fill material, especially translucent orange and red-orange material. The spheres range in color from pale orange to red-orange, overlapping the color range of but never appearing as deeply red as some other vessel fills. In some vessels, it is unclear whether the vessel is densely filled with spheres, filled with blocky grainy material, or a mixture of the two (Fig. 3A). The individual spheres range in size from 6 to 15 µm, but the majority are about 10 µm in diameter. In areas where the spheres are sparse and easily distinguished from one another, their external texture appears bumpy, and appears to be composed of 1–3 µm round subunits. These subunits are best visible when adjusting focus through the depth of the section.

Sections 2018-X1 and 2018-X2 (see Table 1; not figured) were cut transversely in sequence from a 3.5 × 1 mm fragment provided by Dr. Yao. The edges of these slices are gently curved, as they are part of the circular transverse section of another gastralium or small rib fragment. The transverse sections reveal dark-filled osteocyte lacunae with distinct canaliculi, as well as primary osteons. The contents of these osteons’ central canals are not as easy to discern in this view; the variation in textures and colors of vessel fill is not apparent and neither are the spheres.

The petrified wood thin section, 2018-1 (Table 1, Fig. 4), is a square slice with two zones of carbonate preservation surrounding a zone of silicious preservation. The carbonate zones do not preserve histological detail. Within the silicious zone, however, lines of boxy segments each about 20–40 µm long are visible; these are possibly tracheid cells of the xylem. Some of these segments contain dark round structures that range in size from <5 µm to 20 µm and are similar in appearance to the purported blood cells in the associated bone (Fig. 4, arrowheads).

Figure 4 Transmitted light micrographs of petrified wood from thin section 2018-1.

(A) Spheres ranging in size from approximately 5–20 µm are visible at low magnification (blue arrows mark small and large examples). (B) At higher magnification, the spheres appear to be made up of smaller crystals.

b. Raman spectroscopy

Raman spectra acquired from across these specimens are characterized by a complex of peaks around 1,000–1,700 cm−1. Apatite peaks are visible in most of the Raman spectra, with an average position of 962.4 cm−1 (range: 959.8–965 cm−1; Table S1). Spectra collected from the plexiglass slide, epoxy, and superglue around our specimen had no peak overlap with our sample spectra (see Fig. S1).

We did not observe peaks for pyrite (FeS2) or its associated iron oxides, which would occur at wavenumbers below 700 cm−1 (De Faria, Venâncio Silva & De Oliveira, 1997; Vogt, Chattopadhyay & Stolz, 1983). We also did not observe any of the five marker peaks for heme compounds, which all occur above 1,340 cm−1 (Asher, 1981; Schweitzer et al., 1997).

c. Energy dispersive X-ray spectrometry

i. Hitachi TM-3000

The most abundant elements in the bone under EDS are phosphorus and oxygen (Figs. 5D, 5F). The signals from filled vessels differ across the thin section. Some, especially in regions noted under light microscopy to have many spheres, are dominated by aluminum and silicon, indicating the possible presence of clay minerals (Figs. 5B, 5E). At other locations, carbon is concentrated in the vessels and abundant in void spaces and fractures in the bone. Due to the differences in microscopy, we could not positively identify a sphere as identified on a light microscope under SEM. Figures 5B and 5E–5F show that vessel fills differ in composition from the surrounding bone.

Figure 5 Energy Dispersive X-ray Spectroscopy (EDS) data from thin section 2018-L2.

(A) Scanning electron micrograph of an area analyzed with EDS. (B–F) EDS elemental maps of interest, with the mapped element marked in the lower left of each panel. Relative abundance of an element is indicated by color brightness. (G) Spectrum of elements present across the whole analyzed area. (H–I) SEM images of areas of 2018-L2 indicating EDS analysis points of vessel and lacuna fills. (J) EDS results at the point noted as “1” in (H) showing kaolinite filling a vessel and lacuna. Elemental ratios were calculated from atomic percentages.

ii. FEI quanta 600FEG

Using area scans of various exposed vessels on 2018-L1, we found some fills with carbonaceous compositions and others composed of aluminosilicates. The low-chlorine carbon-rich fills were distinguishable from chlorine-rich epoxy infill by their chlorine content. By employing quantitative spot analyses of the aluminosilicate fills, we found aluminum to be, by far, the dominant element, with high magnesium and calcium contents (7.96, 2.85, and 1.25 atomic %, respectively). Based on a magnesium number of 100 (Mg was detected in the spot analyses, but not Fe) and a very low ratio of alkalis to aluminum ((Na+K)/Al = 0.121), we identified the aluminosilicates as kaolinite clay minerals (Page et al., 2008).

d. Time of flight –secondary ion mass spectrometry

The analyzed material is a transversely cut one-mm-thick section of a gastralium fragment in situ in finely laminated mudstone that also contains a round, orange-colored concretion (2018-X3 in Table 1). The laminations of the mudstone dip between the concretion and the bone. This scan is a composite of 21 serial scan segments that together measure 1,500 by 3,500 µm (Figs. 6A–6C). This scan shows a clear difference in composition between the bone and its surrounding matrix, with silicates in the matrix and phosphate in bone, and reveals silicates in the pore spaces of the bone visible at this scale (Fig. 6, compare D to I). Carbonaceous compounds are concentrated in the bone and at the contact between the laminated matrix and concretion but are not abundant within the concretion or matrix (Fig. 6J). Iron and manganese oxides are evenly dispersed through the matrix and nearby concretion. Aluminum is most densely concentrated in the concretion. Outside of the concretion, Al is densest where Si is also present, except in the matrix densest with Cl. Both 35Cl and 37Cl are concentrated in the bone, but also infiltrate the matrix in contact with the bone (Fig. 6F). The concentration of chlorine within the matrix decreases with distance from the bone. Fluorine is abundant throughout the sampled area but is especially concentrated in the bone (Fig. 6H).

Figure 6 Time of Flight Secondary Ion Mass Spectrometry (TOF-SIMS) maps of an in situ bone fragment and the surrounding matrix, specimen 2018-X3.

(A) The cut fragment with cross section of a gastralium, fine lamination in surrounding shale, and an orange-tinted concretion. Examined area indicated by rectangle. (B) Drawing of the scanned area emphasizing borders between bone, sedimentary matrix, and concretion. (C) TOF-SIMS map of total chemical species, consisting of a collage of 21 half-millimeter squares. (D–J) TOF-SIMS maps of six chemical species of interest, with the mapped chemical species noted in upper left of each panel. These six species represent a subset of the total collected species.

Discussion

The bone of Beipiaosaurus inexpectus (IVPP V11559) is highly chemically altered. Apatite in the holotype gastralia has incorporated both fluorite and carbonates, resulting in a Raman peak position at a relatively low wavenumber (ca. 962 cm−1; Fig. 3). This peak has a low wave number despite abundant fluorine in the bone (Fig. 6H), which drives apatite peaks towards higher wavenumbers. Raman peak positions for bioapatite shift higher with the addition of fluorine through diagenetic alteration. This results in a range of Raman shifts from 962 cm−1 in modern bone (Alligator mississippiensis) and fossil bone with little alteration, to 966 cm−1 in highly altered fossil bone and in fluorapatite (Thomas et al., 2011).

The average position for the apatite peaks in our samples is 962.4 cm−1, much closer to unaltered enamel apatite (962 cm−1) than to fluorapatite (966 cm−1). However, TOF-SIMS analysis shows high concentrations of both chlorine and fluorine in Beipiaosaurus bone. There is a gradient in Cl concentration from the bone into surrounding matrix observed with TOF-SIMS (Fig. 6F). The chlorine is likely a component of altered apatite (Keenan, 2016), and the diffusion gradient indicates that exchange between the bone and matrix has occurred, despite the low average wavenumber position of the apatite peak observed with Raman. Notably, carbonate in the bone has the opposite effect of fluoride, shifting the peak towards lower wavenumbers (Li & Pasteris, 2014; Thomas et al., 2011). Many of our apatite peaks are below 962 cm−1, indicating a considerable carbonate influence on the peak position. Comparative analysis of modern and fossil bone specimens using integrated TOF-SIMS and Raman analysis is needed to further quantify the impact of fluorine on apatite Raman peak position.

The prevalence of diagenetic materials in the vascular canals also suggests that the bones of this fossil are chemically altered. Using EDS and Raman, we observed a high carbon content in the bone and some vessel (pore space) fills but could not distinguish specific carbonaceous compounds. Additionally, we observed carbon compounds both at the permeable contact between the concretion and matrix and in bone using TOF-SIMS analysis (Fig. 6J). This implies that at least some of the carbonaceous materials in the pores of the bone may be exogenous, either carried in by fluids during diagenesis or present due to post-burial microbial occupation of pore space (e.g., vascular canals) and fractures throughout the bone and matrix.

We did not identify pyrite in our sample with Raman or EDS, even though 5–20 µm iron oxide pseudomorphs of framboidal pyrite have been observed in siliciclastic and carbonate precipitate laminations in sediments of the Jianshangou Unit in the Sihetun area (Hethke et al., 2013) and in unlaminated black mudstones at the Sihetun locality (Zhang & Sha, 2012). Additionally, pyrite framboids ranging in size from 5–31 µm are found in association with insect and plant fossils of the Jehol biota (Leng & Yang, 2003; Wang et al., 2012).

The complex of Raman peaks at 1,212, 1,296, 1,430, 1,570, and 1,690 cm−1 (Fig. 3) fall in a similar position to peaks identified in previous publications as Raman bands of carbonaceous compounds such as amides (Lee et al., 2017; Morris & Mandair, 2011; Puech et al., 1986) (Fig. 3). A similar peak complex has been observed in other spectra obtained from fossil dinosaur bone of various ages (e.g., Lee et al., 2017; Wang & Yang, 2007). The peaks in this region do not match any of the materials used in processing nor were these materials detected when targeting the specimen, indicating that our spectra are from the sample itself rather than contamination. The peaks do, however, match the expected positions of photoluminescent bands, an artefact in Raman spectroscopy that is produced by rare earth elements in fossil apatite and the surrounding shale matrix (Culka & Jehlička, 2018; Lenz et al., 2015) and we interpret them as such. We recommend that any future Raman spectroscopic analyses of fossil bone carbonaceous compounds or biomolecules target more recently collected specimens (per Pruvost et al., 2007) and employ corrective methods for rare earth element artefact bands (outlined in Culka & Jehlička, 2018).

We propose that the spheres in the vascular canals of B. inexpectus are botryoidal structures consisting of authigenic or diagenetic kaolinite clays (Fig. 7). Although we were unable to specifically target the spheres, the EDS and Raman analyses of the vessel fills recovered no evidence of heme or diagenetic products that would support an identity of fossilized red blood cells.

Figure 7 Schematic diagram showing the inferred diagenesis of B. inexpectus at a microscopic scale.

All scalebars are 100 µm (A) Drawing reconstructed to a histologic view of the sectioned area of bone in life showing red and white blood cells in a vascular canal and lacunae populated by osteocytes. (B) Post-mortem effects of lacustrine burial, cells lyse as water fills the canals. (C) Early diagenetic changes in anoxic sediment. Sulfate reducing microbes (depicted as orange) in the vessels facilitate pyrite formation (light grey). Diagenetic formation of kaolinite (grey) begins. (D) Later diagenesis. Framboidal minerals are altered over time (grey) and entombed within authigenic kaolinite in various habits (beige and dark grey). Cation exchange occurs between the bone apatite and fluids. (E) Transmitted light photomicrograph of a vessel in thin section, as it appears today.

We did, however, observe similar spheres in a petrographic section of petrified wood (Fig. 4) that was collected with IVPP V11559 and stored with the bone fragments (see Table 1). The presence of these spheres in wood contradicts a blood cell hypothesis but is consistent with authigenic mineralization. The most common alternative hypothesis to other purported erythrocytes with a similar appearance to those studied herein is framboidal pyrite (Kaye, Gaugler & Sawlowicz, 2008; Martill & Unwin, 1997). However, we have found no chemical evidence indicative of pyrite or its oxidative products in our samples, aside from iron and manganese oxides (Mg2O− and FeO−) scattered evenly in low abundance throughout the matrix (detected with TOF-SIMS; see Fig. S2).

The clays detected in IVPP V11559’s vascular canals appear to have a variety of habits in addition to spheres. This does not rule out the possibility that former pyrite structures have been replaced by the clays we observed. In fact, kaolinite can form a pseudomorph of pyrite framboids, and the two minerals are often associated (Pollastro, 1981). Many other fossils bear similarly sized spheres that resemble those we observed in this study (Moodie, 1920; Schweitzer & Horner, 1999; Seitz, 1907; Wilby, 1993), though only some of these are hypothesized to be red blood cells. We are inclined to interpret these similar spherical structures as indicators of shared traits of diagenetic history across many fossil localities, reflecting similarities in the taphonomic histories of disparate fossil-bearing environments rather than red blood cell preservation. Individual analysis of these similarly appearing structures would be necessary to ascertain their compositions and thus identities.

Conclusions

On the basis of comprehensive analyses conducted herein, we suggest that the purported blood cells in the bones of Beipiaosaurus inexpectus (IVPP V11559) are authigenic or diagenetic structures composed of kaolinite clay. Even though we were unable to sample the chemical composition of individual spheres directly, clay minerals are concentrated in the pore spaces of the bone, in the same areas of the thin sections where the spheres are most abundant. This suggests that the spheres and other vessel fills are some of the many habits clays can take. The presence of similar small, reddish spheres in the associated petrified wood also weakens the blood cell hypothesis. One factor limiting the certainty of our results is that these spheres, unmistakable when viewed under a light microscope, were not confidently identifiable with SEM.

To further investigate the nature and origin of these spheres, chemical analyses of extracted or etched specimens may allow better association between particular vessel fill structures and exogenous kerogen or clay. A more recently collected specimen may also be more informative for studying biomolecular preservation in fossil bone from the Yixian Formation.

Supplemental Information

Supplemental Information 1 Peak assignment data for entire TOF-SIMS spectrum documenting manual adjustments and assignment criteria

Click here for additional data file.

Supplemental Information 2 Raman spectra with all data points analyzed in this manuscript continuously reveal the pattern exemplified in the in-text figure

Each spectrum represents a reading at 1/10 power unless otherwise specified. A brief description of each point is in the sheet title.

Click here for additional data file.

Supplemental Information 3 Raw raman spectra acquired from epoxy glued to plexiglass slide do not recover potential contaminating peaks

Click here for additional data file.

Supplemental Information 4 The entire TOF_SIMS spectrum shows peaks that correlate to the data file on peak assignment

Click here for additional data file.

Supplemental Information 5 Raman apatite peak positions, Raman spectra for mounting media, and additional TOF-SIMS data respectively

Click here for additional data file.

This work was completed as part of DEK’s master’s thesis. First, we thank Dr. Jinxian Yao at Peking University and Dr. Xu Xing at the IVPP for lending us the fossil materials used in this study. We thank Andrei Dolocan at the University of Texas at Austin for his expertise with TOF-SIMS. Many thanks to Charles Farley, Jing Zhao, and Dr. Robert Bodnar of the Virginia Tech Vibrational Spectroscopy lab for their support. We would especially like to acknowledge Mr. Farley’s exceptional help to researchers throughout his career, including these authors, and congratulate him on his retirement. We are grateful to Dr. Michelle Stocker for providing samples of modern bone. We thank Chunchi Liao and Shiying Wang for their help in the field, expertise on B. inexpectus and the Yixian Formation, and for their friendship and hospitality. We are grateful to Dr. Caitlin Colleary and Dr. Qing Tang for helpful conversation, advice, and support during this project. We thank Drs. Bruce Lieberman, Evan Saitta, Craig Marshall, and two anonymous reviewers for insightful commentary and helpful feedback. This work was primarily completed on the traditional unceded land of the Tutelo and Monacan people. Settlers held enslaved laborers in captivity on this land and used their labor to build the institution in which this study was undertaken.

Additional Information and Declarations

Competing Interests

Author Contributions

Field Study Permissions

Data Availability

The authors declare there are no competing interests.

Dana E. Korneisel conceived and designed the experiments, performed the experiments, analyzed the data, prepared figures and/or tables, authored or reviewed drafts of the paper, and approved the final draft.

Sterling J. Nesbitt and Sarah Werning analyzed the data, authored or reviewed drafts of the paper, and approved the final draft.

Shuhai Xiao conceived and designed the experiments, analyzed the data, authored or reviewed drafts of the paper, and approved the final draft.

The following information was supplied relating to field study approvals (i.e., approving body and any reference numbers):

Specimen observations and materials were provided by Dr. Xu Xing with permission from the Institute of Vertebrate Paleontology and Paleoanthropology in Beijing for destructive sampling.

The following information was supplied regarding data availability:

All Raman spectra as well as the TOF-SIMS spectrum and peak assignment data are available in the Supplemental File.

The specimens used herein are accessioned at the IVPP in Beijing with the holotype under the museum number IVPP-V11559. The slide numbers are 2018-X1, 2018-X2, 2018-L1, 2018-L2, 2018-L3, 2018-L4, 2018-L5, HO-9601, HO-9602, LJ98B-1, and LJ98B-4, 2018-1, and 2018-X3. The modern alligator bone sample is from un-accessioned material stored in the Virginia Tech osteology collection.

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
