# Peer review of "Putative fossil blood cells reinterpreted as diagenetic structures"

_PeerJ, doi:10.7717/peerj.12651_

## Round 0.1 · original submission · Minor Revisions

I think this is an important paper and I believe it deserves to be published. I've got some small scale suggested revisions that I think should be made before it is accepted. The first relate to the fact that I solicited a reviewer with expertise in the area of using Raman to study soft-bodied fossils and they have made a number of comments and suggestions which I think will be helpful so I encourage you to consider those.

The second point is that as far as I am aware there is only one paper that has used the Raman technique to analyze soft-bodied tissue preservation in the fossil record and given that is exactly what you are focusing on here that paper should be cited. In particular, I would cite: Marshall, A. Olcott, R. L. Wehrbein, B. S. Lieberman, and C. P. Marshall. 2012. Raman spectroscopic investigations of Burgess Shale-type preservation: a new way forward. Palaios 27:288-292.

when you get to the part of your text where you introduce Raman methods and their applicability to this particular problem.

The final suggestion I would make is that there are a couple of places where the text takes a rather strong, definitive approach to the notion that the conclusions refute the possibility that red blood cells might be preserved. I think it would be appropriate to soften the tone in these places, recognizing that you are only providing data supporting or refuting a hypothesis. That is to say, you have provided useful data, and they are consistent with your interpretation. I think that would be an accurate characterization of your (and any) study and a better way to frame the problem.

If you have any questions about any of these requested revisions feel free to reach out to me. With these changes the paper will be good to go for acceptance in my book.

·

Basic reporting

No weaknesses noted. The manuscript is clearly written in professional unambiguous language. The manuscript cites all the relevant literature and provides sufficient background behind the problem being addressed.

Experimental design

The only weakness noted in the Materials and Methods section is minor descriptions/discussion with respect to the Raman spectroscopy section. These are listed here below:

1) Why was the laser spot size 8 micrometers? Was the laser defocussed to avoid laser induced heating on the sample? To this end, what magnitude objective was used?

2) Is the 785 nm laser used in this work a diode laser?

3) What algorithm was used for baseline correction in the acquired Raman spectra?

4) What is the spectral resolution?

Validity of the findings

The only weakness noted in this manuscript is the interpretation of the Raman spectra, with respect to the assignment of the bands in the region of ca. 1000 to 2000 cm-1. The manuscript assigns these bands to carbonaceous compounds and potentially amides. However, the authors have confused Raman bands with photoluminescence (PL) bands. Fossilized bones and shales, in particular, usually contain notable quantities of REEs. And these types of samples when excited by a NIR laser such as a 785nm excitation wavelength (presumbly a diode laser) typically contain REEs with 4f electronic configuration in which illumination excites the sample's valence electrons, and consequently, PL emissions are then generated through the release of energy in (radiative) electronic transitions. This results in PL bands that are similar in FWHM than those of Raman bands arising from organic compounds, which these PL bands occur between 1000 to 2000 cm-1 in the spectrum when excited by NIR lasers. In sum, these REE-related emissions may easily be mistaken as Raman bands.

Additional comments

Apart from some of the methodology missing and the incorrect interpretation of the Raman spectra, this manuscript is almost ready for publication. I would recommend minor revision.

---

## Round 0.2 · accepted · Accept

The authors did an excellent job addressing my suggestions and those from the reviewer. In my judgement the paper should be accepted.